# Remote Auscultation of Heart and Lungs as an Acceptable Alternative to Legacy Measures in Quarantined COVID-19 Patients—Prospective Evaluation of 250 Examinations

**DOI:** 10.3390/s22093165

**Published:** 2022-04-20

**Authors:** Or Haskel, Edward Itelman, Eyal Zilber, Galia Barkai, Gad Segal

**Affiliations:** 1Sackler Faculty of Medicine, Tel-Aviv University, Tel-Aviv 69978, Israel; orhaskel@gmail.com (O.H.); edward.itelman@sheba.health.gov.il (E.I.); eyal.zilber@sheba.health.gov.il (E.Z.); galia.barkai@sheba.health.gov.il (G.B.); 2Internal Medicine “I”, Sheba Medical Center, Ramat-Gan 52621, Israel; 3Beyond, Virtual Hospital, Sheba Medical Center, Ramat-Gan 52621, Israel

**Keywords:** tele-medicine, physical examination, medical device, COVID-19, tytocare, quarantine, digital health

## Abstract

The COVID-19 pandemic accelerated the assimilation of telemedicine platforms into medical practice. Nevertheless, research-based evidence in this field is still accumulating. This was a prospective, cross-sectional comparative assessment of a remote physical examination device used mainly for heart and lung digital auscultation. We analyzed usage patterns, user (physician) subjective appreciation and compared it to legacy measures. Eighteen physicians (median age 36 years (IQR 32–45): two interns, seven residents and nine senior physicians; eleven internists, five geriatricians and two pediatricians) executed over 250 remote physical examinations. Their median work duration with quarantined patients was 60 days (IQR 45–60). The median number of patients examined by a single physician was 17 (IQR 10–34). Regarding overall estimation, all participants tended to prefer the remote examination in the setting of quarantined patients (median 6, IQR 3.75–8), while no statistically significant difference was demonstrated compared to the indifference value (*p* = 0.122). Internists preferred tele-medical examination over non-internists, with significant differences between groups regarding heart auscultation, (median 7, (IQR 3–7) vs. median 2, (IQR 1–5, respectively)), *p* = 0.044. In the setting of quarantined patients, from the physicians’ perspective, a digital platform for remote auscultation of heart and lungs was considered as an acceptable alternative to legacy measures.

## 1. Introduction

The Coronavirus Disease 2019 (COVID-19) pandemic catalyzed rapid adoption of telehealth and transformed healthcare delivery at a breathtaking pace [1,2,3]. Among its most notorious viral characteristics, this new corona virus has a very high infectivity rate [4], necessitating adoption of new technologies for patients’ diagnosis and therapy—in terms of total physical patients’ isolation [5,6]. In such circumstances, that were the common practice at the outbreak of this pandemical disease, patients’ examination becomes challenging. At the Sheba Medical Center, a tertiary hospital in Israel, several departments were converted to provide fully quarantined medical care for COVID-19 patients, including several internal, geriatric, pediatric departments and ICU units. Dedicated equipment was installed for provision of monitoring and communication as well as practicing remote physical examination. Preliminary data regarding similar patients’ cohorts was published earlier [7].

This was a new mission for all involved medical and nursing personnel, necessitating rapid adaptation of telemedicine technologies, of which the TytoCare^®^ system was of paramount importance, enabling remote physical examination of COVID-19 patients. The TytoCare^®^ system is a digital platform designed for remote physical examination including a stethoscope for heart and lungs’ auscultation, a digital otoscope for visualization of the tympanic membrane, a digital thermometer, digital oximeter and a tongue depressor enabling visual examination of the pharynx [8]. Transmitted data and visuals are recorded and a video conference is available as well. Figure 1 displays the system components: a digital monitor presenting self-use instructions (in case the patient places the device for the use of a remote doctor), and the various aforementioned components.

The current work describes the user (physicians) experience and characters of this unique tele-physical examination tool under strict isolation measures. Using a demographic and informative questionnaire, we evaluated and quantified usage characteristics. Over a four-month period, the user experience was tested and compared to standard physical examination, and the findings analyzed and discussed.

## 2. Methods

### 2.1. Study Design and Population

After approval by an institutional ethics committee (Chaim Sheba Medical Center approval number: 7255-20-SMC), we did a cross sectional study of user experience using questionnaires collected from physicians treating COVID-19 patients at internal, geriatric and pediatric departments during the initial phase of the pandemic in Israel, between March–July 2020. The questionnaires were designed together with our epidemiology consultant as simple, Likert-based questionnaires following the relevant literature recommended principles [9]. Thirty physicians were included. Informed consent was an integral part of the physicians’ questionnaires (as approved by the IRB). We collected twenty-five questionnaires and analyzed them to ensure compliance with exclusion and inclusion criteria. Following this, only 18 questionnaires were included in the analysis (Figure 2). A detailed list of dependent and non-dependent variables appears in the results section.

All remote physical examinations were done with video guidance of the physician while the patients themselves were applying the device (for pharyngeal inspection, heart auscultation and frontal lung auscultation and asking their closest fellow patients (in most cases friends and/or family) help while lung auscultation was done from the back.

### 2.2. Statistical Analysis

We described categorical variables as frequency and percentage. Continuous variables were evaluated for normal distribution and reported as median and IQR. In the Visual Analog Scale (VAS) questionnaires, we considered a value of five as indifferent between the two means of testing. A one-sample Wilcoxon rank test used to compare the score to the indifference value. Spearman’s rank coefficient used to evaluate the relationship between continuous variables. The Mann–Whitney U test used to compare continuous variables between categories. The Fisher Exact test was used to compare differences between categorical variables. D Cohen’s effect size was calculated. We considered small, medium, and large differences as greater than 0.2, 0.5, and 0.8, respectively. The purpose of calculating the effect size was to confront the small sample size, which by nature tends to demonstrate the absence of difference between groups (the larger D Cohen’s effect size, the greater the likelihood that if a larger group of subjects were available, statistically significant differences would have been obtained [10]). All statistical tests were two-sided and we considered *p* value < 0.05 as statistically significant. SPSS software used for statistical analysis (IBM SPSS for Windows, version 24, IBM CORP, Armonk, NY, USA, 2016)

## 3. Results

The initial review included satisfaction questionnaires from physicians who conducted more than 250 Tele-examinations. Eighteen questionnaires were eligible for analysis. Eleven male (61%) and seven female physicians (39%) were included. Their median age was 36 years (IQR 32–45). The median years of experience since graduation from schools of medicine was 6 years (IQR 3–13). In terms of ranking, two were interns, seven residents, and nine senior physicians. We obtained most questionnaires from internists, five from geriatric physicians, and two pediatricians. Physician characteristics are presented in Table 1.

We quantified and compared the volume of use of standard physical examinations conducted in full contact environment prior to COVID-19, versus examinations conducted within quarantine wards using TytoCare^®^ system. Within isolated wards the median work duration (days) for physicians was sixty days (IQR 45–60). The median number of patients examined using the TytoCare^®^ system by a single physician was 17 (IQR 10–34). The median hospitalization length was 8 days (IQR 7–10), with a single daily examination being executed on average per patient. All participating physicians indicated performing daily lung auscultation. Ninety percent (IQR 80–100) also conduced heart auscultation. Only 5% (IQR 0–23) used the option for pharyngeal visualization. Ear canal examinations were not performed at all. In contrast, prior to the pandemic the median amount of standard physical examinations performed daily, in patients presenting with respiratory symptoms in a non-isolated ward, using legacy measures was five (IQR 3–5). Daily lung auscultations were five (IQR 3–5), heart auscultations were five (IQR 3–5), and pharynx visualization was one (IQR 1–5).

User (physician) experience outcomes were compared between modalities, using a nine-values VAS, with the score of 5 marking indifference between modalities while 1 indicates a major preference in favor of the standard physical examination and 9 indicated a major preference in favor of the TytoCare^®^ system. We measured statistical significance, with the null hypothesis of no significant preference of one test over the other (non-inferiority). Results are summarized in Table 2.

### 3.1. Comparison in the Domains of Convenience and Quality, as Perceived by the Operators

Regarding lungs examination, in terms of convenience (as perceived by the participating physicians) the responders preferred the traditional examination (median 2, IQR 1–7). No statistically significant difference was demonstrated when compared to indifference value, *p* = 0.065. For quality, there was considerable indifference between examinations (median 5, IQR 1.75–8); *p* = 0.826. Regarding heart auscultation, in terms of convenience indifference was evident in comparison between examinations (median 5, IQR 1–7.25); *p* = 0.469. For perceived quality of heart auscultation there was also indifference in comparison (median 5, IQR 3–7.25); *p* = 0.917. Regarding pharynx visualization convenience, there was a preference for the TytoCare^®^ system (median 7, IQR 5–9), with statistical significance (*p* = 0.009). For the pharynx Inspection perceived quality there was also a preference for the TytoCare^®^ system (median 7, IQR 5–9); *p* = 0.038.

## 3.2. Comparison in the Domains of Patients’ Compliance and Overall Estimation

Responders presented indifference in comparisons for patient compliance (median 5, 5–6.25 IQR). No statistically significant difference was demonstrated compared to indifference value, (*p* = 0.259). Regarding overall estimation, responders tended to prefer the TytoCare^®^ system (median 6, IQR 3.75–8), *p* = 0.122. As for recommending use of tele-physical examination equipment in a non-isolated environment, responders tended not to recommend tele-physical examination in a non-isolated environment (median 3, IQR 2–5.25), *p* = 0.122. Regarding the question whether the telemedicine device provides an adequate solution for the standard physical examination in isolation conditions, 72% of the respondents answered positively (*n* = 13).

A subgroup division was made regarding the years of experience of physicians (summarized in Table 3). The cut-off was set at over 4 years of experience, and below. The median score given to the VAS was calculated, as well as statistical significance (the null hypothesis was no difference). Effect size D was calculated to analyze differences between small sample groups. The senior group ranked higher for most parameters, but without significance. Regarding pharynx visualization convenience, the senior group ranked higher. Effect size D indicated of large difference between groups (median score of 8.5 vs. five, D = 0.89).

We made an additional comparison between internal ward physicians versus others, which tended to prefer tele-medical approach without statistical significance. Regarding heart auscultation convenience using tele-medicine, internists had a significant higher preference for the TytoCare^®^ system (median score 7 vs. 2, respectively, *p* < 0.05). Ninety one percent of internists answered positively regarding the question whether tele—examination provided an adequate solution under quarantine environment. Among non-internists, only 43% answered positively to this question. This difference was statistically significant (*p* = 0.047). The data are summarized in Table 4 and Table 5.

## 4. Discussion

The term “telemedicine”, as defined by the American Telehealth Association (ATA), is “the use of the transfer of medical information from one site to another by electronic means of communication in order to improve a patient’s medical condition.”. Until the beginning of the 21st century, the use of “telemedicine” was common, especially in indications in which the barrier of availability and distance had to be overcome: Its first application goes back in the USA civil war, using telegraph machines to transmit casualty list and order medical instruments [11]. In the maritime environment, as a lesson learned from the Titanic tragedy in 1912, sailors could ask for emergency medical consultation using the “Medico” procedure over radio communication [12]. In modern times one of the earliest efforts to overcome time and distance barrier was part of NASA space flight program concerning the physiological monitoring of astronauts [13]. With regard to medical research, the very first time “Tele-medicine” was mentioned in literature was only in 1974 [14]. The computing and internet revolution during the past 20 years opened a variety of applications to be implemented as part of healthcare systems, such as general practitioners’ remote visits, oncology follow-ups, chronic patient complex care and pediatric visits. This new paradigm was constantly growing, but the volume of tele-medicine as part of the routine daily care system was not extensive until recently. The process of incorporating tele-medicine into healthcare general systems has taken a significant turn worldwide during the first quarter of 2020, when the COVID-19 pandemic emerged. Social distancing, along with high infectivity rate increased the need for various uses of telemedicine applications and allowed medical personal to assimilate these uses both as part of the treatment of isolated patients in the inpatient wards, but also for non-COVID-19 patients, as part of ambulatory care. For example, by April 2020 nearly half (43.5%) of the US Medicare primary care visits were already provided through telehealth compared with less than one percent (0.1%) in February, before the pandemic emerged [15,16].

Physical examination has always been a cornerstone of diagnostic and medical care. Different disciplines apply it differently, but observation of general appearance, followed by auscultation to the heart and lungs using a stethoscope are common for all. Providing telemedicine without direct physical contact, considered by clinicians as the “holy grail” of the medical encounter, therefore raises fundamental challenges. Clinical attempts to perform tele-physical examination including the four basics instruments: observation, palpation, percussion, and auscultation remotely were scarce, and the medical literature is void of such studies, especially for inpatients. Our study summarizes a unique, preliminary experience that was forced on our medical personnel, in light of the need for physically examine quarantined patients. We therefore found an urgent need to describe, assess and compare the usability and convenience (as subjectively conceived by the users/physicians) of this system with the conservative measures already known to physicians. Taking int account the physicians’ specialty, gender, professional experience was obviously necessary.

In this study, we reviewed the volume of use and user (physician) experience in the setting of inpatient, COVID-19 management. As stated earlier, we compared it to legacy approach. We used the TytoCare^®^ system for the purpose of remote cardio-pulmonary auscultation. In contrast to previous attempts of applying a wireless stethoscope in the setting of quarantined, COVID-19 patients [17,18], the current study describes the TytoCare^®^ remote auscultation probe, as a part of a more holistic tele-physical examination tool.

The current pandemic introduces a conflict for physicians: on the one hand, the desire to provide best medical care to patients and on the other hand the risk of being infected. The median duration of hospitalization length in quarantine wards (8 days), which is longer than the average duration in an internal ward (5.5 days) further sharpen this conflict. As expected in the setting of a severe respiratory disease, all examinations included lung auscultation, the vast majority (90%) included heart auscultation, and a minority (5%) included pharynx visualization.

During the COVID-19 pandemic, most of the telemedicine applications were applied for ambulatory patients in the setting of home care [19]. Previous descriptions of tele-physical examination in the fields of rheumatology, orthopedics and neurology were published with clinician’s relaying on various techniques, including analyzing the patient’s video images and calculating joint angles. Another discipline in which tele-physical examination gained significant development is otolaryngology, mainly for pediatrics [20,21,22,23]. The use of the TytoCare^®^ system by ambulatory pediatricians was also previously reported to be doubled during the COVID-19 pandemic in Israel [24].

### Comparison of Tele-Medical Assessment and Legacy Measures

The comparative results show that the tele-examination was perceived, by most physicians’ and for the majority of tests/procedures, as an acceptable alternative to standard measures. Regarding inspection of the pharynx, in contrast to past attempts, the TytoCare^®^ platform provided a successful solution [25]. The perceived quality of care within a quarantine ward is highly influenced by telemedicine user’s (physician) experience, potentially overcoming the physical distance between the therapist and the patient. In the current work, we focused on both examination convenience and the physicians’ perceived quality of inspection. It appears that pulmonary auscultation was the only component in which there was a preference to legacy measures. Heart auscultation was perceived as non-inferior, whereas pharynx evaluation was experienced as superior to standard examination. Our results show that the TytoCare^®^ user interface was designed well, since despite the limitations of distance and isolation, there was indifference between the examinations with regard to patient compliance. In overall estimation comparison between modalities, the respondents tended to prefer the tele-physical examination in the setting of need for distant patients’ physical examination (median score = 6, higher than the indifference score = 5), but when they were asked if they would recommend using the platform in a full contact ward, the median score was only 3, showing that they apparently did not support this statement. We hypothesize that this preference should be considered as a normal first step towards an adoption of a novel technology. The foreign/new environment gave the physicians a certain justification to “get out” of the convenient, known, legacy technologies. It is only natural for them to postpone the adoption of novelties into their old environment. Such delays in adoption of technologies are known and discussed elsewhere in the literature, both in terms of technologies’ assimilation methods [26] and way of addressing skepticism amongst physicians [26].

## 5. Conclusions

Our unique and primary experience shows that usage of TytoCare^®^ system for remote physical examination provided a positive experience, as reflected by subjective satisfaction reviews of a heterogenous, albeit small group of physicians. The fact that this disruptive technology is not experienced as inferior to the standard physical examination has a great significance to the medical community. We do not suggest “non inferiority” as this term is used in prospective, comparative studies but rather present our results as indicating the fact that this technology is considered as a good alternative to legacy measures in the setting of “the remote patient”. In addition, it can be interpreted that use of a tele-physical examination platform has the potential to increase the level of medical certainty. Every future technology will have to face all barriers of instruction, assimilation, training and verification of usability in real-life experiences prior to large-scale adoption. Further exploration and development of usage guidelines of the platform described and similar are needed. Additionally, further qualitative research is needed in order to better understand physicians’ preferences for technological alternatives they are presented with.

## 6. Study Limitations

This is a descriptive cross-sectional study conducted using satisfaction questionnaires, with all the limitations implied in this outline. The data projects the subjective assessment of the discussed tools. We managed to maximize the external validity of our findings, limited due to the small size of the study population using the D Cohen’s Effect size). Further studies need to be carried out on both larger population multicenter studies.

## 7. Full Disclosure

Researchers have no commercial relationship with the device manufacturer. The device used in this study was not donated or transferred for use in research. The commercial firm/manufacturer of Tyto was totally unaware to this research and had no connection whatsoever to the writing of the manuscript. All these restrictions were presented to the institutional ethics committee.

## Figures and Tables

**Figure 1 sensors-22-03165-f001:**
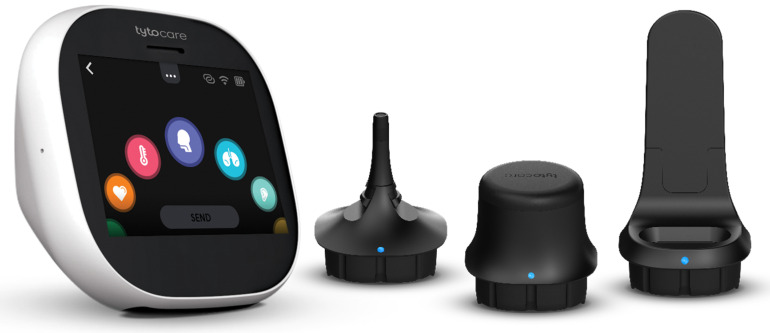
The TytoCare^®^ System.

**Figure 2 sensors-22-03165-f002:**
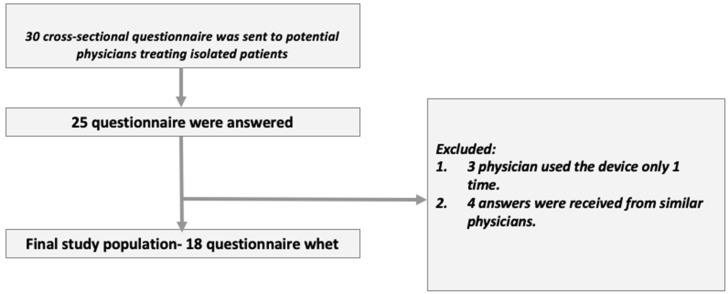
Study flow.

**Table 1 sensors-22-03165-t001:** Characteristics of physicians participating in this study.

Overall	*n* = 18
Male	11 (61%)
Age	36 (32, 45)
Physician ward	
Internal	11 (61%)
Pediatric	2 (11%)
Geriatric	5 (28%)
Physician ranking	
Internship	2 (11%)
Residence	7 (39%)
Senior	9 (50%)
Years of experience since graduation	6 (3, 13)

**Table 2 sensors-22-03165-t002:** Physicians’ differential preferences for examination modalities: score of 5 marking indifference between modalities while 1 indicates a major preference in favor of the standard physical examination and 9 indicated a major preference in favor of the TytoCare^®^ system.

Parameter	Median (IQR]	*p*-Value
Lung auscultation convenience	2 (1–7)	0.065
Lung auscultation Inspection quality	5 (1.75–8)	0.826
Heart auscultation convenience	5 (1–7.25)	0.469
Heart auscultation Inspection quality	5 (3–7.25)	0.917
Pharynx visualization convenience	7 (5–9)	0.009
Pharynx visualization Inspection quality	7 (5–9)	0.038
Patient compliance comparison between methods	5 (5–6.25)	0.259
Overall estimation	6 (3.75–8)	0.122
Recommendation in a non-isolated environment **	3 (2–5.25)	0.122

** Score = 1, will not recommend at all. Score = 9 would highly recommend.

**Table 3 sensors-22-03165-t003:** Physicians’ differential preferences for examination modalities according to seniority: score of 5 marking indifference between modalities while 1 indicates a major preference in favor of the standard physical examination and 9 indicated a major preference in favor of the TytoCare^®^ system.

Parameter	Years of Experience <5; *n* = 8 (Median, (IQR))	Years of Experience ≥5; *n* = 10 (Median, (IQR))	*p*-Value	Effect Size D (Cohen’s)
Lung auscultation convenience	2.5 (1–7)	3.5 (1–7.25)	0.965	0.07
Lung auscultation Inspection quality	3.5 (1.25–8.75)	6 (3.25–7.25)	0.897	0.15
Heart auscultation convenience	5 (2.25–7.75)	3.5 (1–7.5)	0.573	0.25
Heart auscultation Inspection quality	4.5 (3–7.75)	5.5 (3.25–7.25)	0.897	0.03
Pharynx visualization convenience	5 (4–8)	8.5 (5.5–9)	0.121	0.89
Pharynx visualization Inspection quality	5 (4–8)	7 (5.5–9)	0.281	0.68
Patient compliance	5 (4.25–6.75)	5 (5–6.25)	0.762	0.30
Overall estimation	4.5 (3–8)	6 (5–8.25)	0.315	0.41
Recommendation using tele-physical examination in a non-isolated environment **	4 (2.25–8)	2.5 (1.75–5.25)	0.460	0.48

** Score = 1, will not recommend at all. Score = 9 would highly recommend.

**Table 4 sensors-22-03165-t004:** Volume and characteristics of use with regard to physicians’ specialization.

Parameter	Pediatric and Geriatric Physicians, *n* = 7	Internal Ward Physicians *n* = 11	*p*-Value	Effect Size D (Cohen’s)
Age	45 (33–46)	35 (32–41)	0.21	0.65
Years of Experience	12 (2–20)	4 (3–12)	0.37	0.72
Patients examined using TytoCare^®^ system	15 (3–20)	20 (10–45)	0.42	0.23
Average hospitalization length (days)	7 (7–10)	10 (7–12)	0.15	0.6
Tele-physical examinations performed (average/patient/day)	1 (1–2)	1 (1–1)	0.47	0.59
Days in quarantine ward	45 (14–110)	60 (50–60)	0.42	0.16
Out of telemedicine visits, how many included tele-examination (%)	50 (15–80)	20 (15–55)	0.47	0.61
Lung auscultation (% of tele-examinations)	100 (95–100)	100(100–100)	0.96	0.26
Heart auscultation (% of tele-examinations)	95 (88–100)	90 (80–100)	0.25	0.58
Pharynx visualization (% of tele-examinations)	5 (0–5)	10 (0–30)	0.42	0.65
Ear canal visualization (% of tele-examinations)	0	0	0.65	0.61
Skin visualization (% of tele-examinations)	0	0 (0–2)	0.93	0.24
In what percentage of examinations, did the tele-device had a clinical impact (%)	30 (25–80)	30 (20–70)	0.93	0.03
Number of sessions required to gain confidence in Tele- examination	3 (1–5)	3 (2–5)	0.79	0.35

**Table 5 sensors-22-03165-t005:** Tele-physical examination user (physician) experience evaluation according to physician specialization: score of 5 marking indifference between modalities while 1 indicates a major preference in favor of the standard physical examination and 9 indicated a major preference in favor of the TytoCare^®^ system.

	Pediatric and Geriatric Physicians, *n* = 7	Internal Ward Physicians *n* = 11	*p*-Value	Effect Size D (Cohen’s)
Does Tele- Device Provide an adequate solution?	43% (*n* = 3)	91% (*n* = 10)	0.04	
Lung auscultation convenience—comparison	1 (−13)	7 (1–8)	0.06	>0.99
Lung auscultation perceived quality—comparison	4 (1–6)	6 (3–8)	0.12	0.76
Heart auscultation convenience—comparison	2 (1–5)	7 (3–9)	0.04	>0.99
Heart auscultation perceived quality—comparison	4 (1–5)	6 (3–8)	0.15	0.77
Pharynx visualization convenience—comparison	6.5 (4.75–9)	7 (5–9)	0.86	0.10
Pharynx auscultation perceived quality—comparison	6 (4.75–7)	8 (5–9)	0.14	0.49
Patient compliance—comparison	5 (5–5)	5 (5–7)	0.28	0.72
Overall estimation of tele-examination compared to standard examination	5 (3–6)	6 (5–9)	0.21	0.68
Recommendation to other practitioners of using TytoCare in a non-isolated environment **	2 (1–5)	4 (2–7)	0.24	0.61

** Score = 1, will not recommend at all, Score = 9 would highly recommend.

## Data Availability

Study data are available with the authors upon individual requests.

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
