# Peer review of "Remote Auscultation of Heart and Lungs as an Acceptable Alternative to Legacy Measures in Quarantined COVID-19 Patients—Prospective Evaluation of 250 Examinations"

_sensors, 2022, doi:10.3390/s22093165_

Round 1

Reviewer 1 Report

The presented work describes an interesting and valuable investigation regarding the use of remote devices for the auscultation, carried out during the pandemic, where many patients were forced into quarantined wards.

Despite the overall importance of such investigation, which indeed deserves to be published, there are some major weakness in the way this investigation is presented that must be addressed adequately.

The very first one is that the title appears to be a bit too strong sentence, especially talking about “measures”, which do not appear to be supported by an adequate body of results, since the “measures” considered in the work refer only to questionnaires administered to some clinicians. I hence recommend the authors to review the whole work, being more careful about possible overemphasized sentences which may result misleading. This in particular since the results are somewhat controversial: as an example, the abstract reports “all participants tended to prefer the remote auscultation”, while along the text it appears that “responders tended not to recommend tele-physical examination in a non-isolated environment”.  Another controversial point, that deserves to be further discussed and possibly explained is that pulmonary auscultation seems to be “the only component in which there was a preference to legacy systems”, given the fundamental role of such in the au

Given the above issues, and the fact that there is not a real quantitative comparison between the signals acquired by the tele-auscultation system and the legacy ones, sentences like “the use of tele-physical examination platform has increased the level of medical certainty” appear to be somewhat not fully justified.

Moreover, there are some crucial points:

  1. Please describe how the measurements were performed (if patients alone or with the help of nurses, ward clinicians, OT or other clinical operators)
  2. In the paper the term “user” seems to be referred to clinicians, while those who are “using” these devices are patients (and/or possibly nurses or other ward operators – see previous issue). Please address properly and review accordingly. This holds particularly true when it comes to the “usability” of such systems, that must be declined either in the operators’ side and the patients’ one.
  3. The term “convenience” is used, but it would be very important to further clarify its meaning
  4. It is unclear if there is a uneven distribution of the daily auscultations among the professionals involved. It would instead be interesting to see if the operators using more often the system are those who appreciate it or rather they are using it only because they are forced to.
  5. Looking at table 4, it seems that, beside the different clinical areas, there may be some differences due to the age of the clinicians. Please address this issue properly
  6. I suggest the author to reorganize the “discussion” section, by moving the initial part describing telemedicine and auscultation in the “introduction” section, and rather further discussing about why the interviewed clinicians expressed an interest in using these devices in a confined ward and do not seem to recommend it in normal one.
  7. Please address properly and extensively the possible conflicts of interests, since describing a commercial product, the final statement about the lack of “commercial relationships” does not appear to be enough.
  8. Please discuss a bit more about the comparison of this new tele solutions with respect to the legacy ones, in particular facing those features regarding usability, training, reliability, etc.
  9. Please introduce some reasoning about the potential use of such interesting telemedicine solutions in confined and non confined wards and at home, highlighting the possible aspects that may support a wider adoption of such systems, whenever such systems proved to have same or better quality with respect to the legacy ones.
  10. For the sake of the discussion, it may be useful to directly present the questionnaire.

Author Response

Dear Reviewer, 

On behalf of all authors, I thank you for your attention and professionalism in reviewing our work. I have no doubt that the publication will become much better thanks to your work.

Prof. Gadi Segal, MD

Comment 1:

The very first one is that the title appears to be a bit too strong sentence, especially talking about “measures”, which do not appear to be supported by an adequate body of results, since the “measures” considered in the work refer only to questionnaires administered to some clinicians. I hence recommend the authors to review the whole work, being more careful about possible overemphasized sentences which may result misleading.

Answer 1:

Indeed, we rephrased the article title, making our statements more prudent.

Comment 2:

This in particular since the results are somewhat controversial: as an example, the abstract reports “all participants tended to prefer the remote auscultation”, while along the text it appears that “responders tended not to recommend tele-physical examination in a non-isolated environment”.  

Answer 2:

The sentence in the abstract was changed accordingly.

Comment 3:

Another controversial point, that deserves to be further discussed and possibly explained is that pulmonary auscultation seems to be “the only component in which there was a preference to legacy systems”, given the fundamental role of such in the au

Given the above issues, and the fact that there is not a real quantitative comparison between the signals acquired by the tele-auscultation system and the legacy ones, sentences like “the use of tele-physical examination platform has increased the level of medical certainty” appear to be somewhat not fully justified.

Answer 3:

The authors have gone over the whole manuscript and changes the appropriate places according to this important and correct comment.

Comment 4:

Please describe how the measurements were performed (if patients alone or with the help of nurses, ward clinicians, OT or other clinical operators)

Answer 4:

A description was added to section 2.1

Comment 5:

In the paper the term “user” seems to be referred to clinicians, while those who are “using” these devices are patients (and/or possibly nurses or other ward operators – see previous issue). Please address properly and review accordingly. This holds particularly true when it comes to the “usability” of such systems, that must be declined either in the operators’ side and the patients’ one.

Answer 5:

We went over the whole manuscript and made sure that the term “user” will be used only for the physicians.

Comment 6:

The term “convenience” is used, but it would be very important to further clarify its meaning

Answer 6:

We added a more detailed description to what we meant, as a subjective parameter on behalf of the users/physicians.

Comment 7:

It is unclear if there is a uneven distribution of the daily auscultations among the professionals involved. It would instead be interesting to see if the operators using more often the system are those who appreciate it or rather, they are using it only because they are forced to.

Answer 7:

We believe that answering this question is important and appropriate but should be considered as a task for our next studies with a significant number of users/physicians.

Comment 8:

Looking at table 4, it seems that, beside the different clinical areas, there may be some differences due to the age of the clinicians. Please address this issue properly

Answer 8:

The difference in “years of experience” had no statistical significance and therefore, we did not address the issue.

Comment 9:

I suggest the author to reorganize the “discussion” section, by moving the initial part describing telemedicine and auscultation in the “introduction” section, and rather further discussing about why the interviewed clinicians expressed an interest in using these devices in a confined ward and do not seem to recommend it in normal one.

Answer 9:

A new paragraph addressing this important and interesting question was added to the “discussion”.

Comment 10:

Please address properly and extensively the possible conflicts of interests, since describing a commercial product, the final statement about the lack of “commercial relationships” does not appear to be enough.

Answer 10:

The “full disclosure” paragraph was further detailed and elaborated in order to reflect the reality of “no association” with any commercial firm.

Comment 11:

Please discuss a bit more about the comparison of this new tele solutions with respect to the legacy ones, in particular facing those features regarding usability, training, reliability, etc.

Answer 11:

A paragraph relating to these issues was added to the conclusions. Since we did not apply other tele-solutions, we refrain from suggesting anything relating to other technologies.  

Comment 12:

Please introduce some reasoning about the potential use of such interesting telemedicine solutions in confined and non confined wards and at home, highlighting the possible aspects that may support a wider adoption of such systems, whenever such systems proved to have same or better quality with respect to the legacy ones.

Answer 12:

In this manuscript we detail our experience with one tele-medicine technology. This is not a review of the field. I am afraid that such review will be rejected by others. Please advise me, as editors, to what extent should such a discussion take place in this publication. We will proceed according to your recommendations.

Comment 13:

For the sake of the discussion, it may be useful to directly present the questionnaire

Answer 13:

The questionnaire was not in English. If deemed essential, we could translate to English and add as a supplementary material.

Reviewer 2 Report

The fatal flaw in this manuscript is the use of the term 'non inferior' in the title and descriptors.  non-inferiority is a very specific statistical state where a (usually quite large) sample of subjects is tested and found with confidence to be equivalent to a different group.  This study has a tiny sample size and cannot make that claim.  If you strip away the 'non-inferiority' language, the study reveals that two small groups of people were compared and there was no difference.  This is a non-event.  it could be that the authors simply did not study enough people to find the differences that exist (risk of type II error).

What is left is a series of not very interesting descriptive statistics.

Author Response

Dear Reviewer, 

On behalf of all authors, I thank you for your attention and professionalism in reviewing our work. I have no doubt that the publication will become much better thanks to your work.

Prof. Gadi Segal, MD

Comment 1:

The fatal flaw in this manuscript is the use of the term 'non inferior' in the title and descriptors.  non-inferiority is a very specific statistical state where a (usually quite large) sample of subjects is tested and found with confidence to be equivalent to a different group.  This study has a tiny sample size and cannot make that claim.  If you strip away the 'non-inferiority' language, the study reveals that two small groups of people were compared and there was no difference.  This is a non-event.  it could be that the authors simply did not study enough people to find the differences that exist (risk of type II error).

What is left is a series of not very interesting descriptive statistics.

Answer 1:

The term “non inferior” was taken out of the manuscript along with every other element that could potentially infer statistical non-inferiority. Nevertheless, we addressed the issue of “small groups” in our professional data analysis. We worked closely with a senior epidemiologist and our statistical presentation is both clear and sound: this is the reason for the addition of the “Cohen’s effect size” calculations. Please re-consider your comment in light of this particular statistical aspect of our work.

Reviewer 3 Report

Dear editors,

Thank you for the opportunity to review this article. I found this paper very interesting, and I think that it is a pilot study that can be helpful for other authors and future research. Nevertheless, I believe that some aspects could perhaps be revised to improve the readability of the text. These are the following:

The authors describe that «we did a cross sectional study of user experience using questionnaires collected from physicians treating COVID-19 patients at Internal». I would invite them to make a better description of the questionnaires. If they were previously validated, or if they required translation or transcultural adaptation.

The authors describe that «We collected twenty-five questionnaires and analyzed them to ensure compliance with exclusion and inclusion criteria. Following this, only 18 questionnaires were included in the analysis». This number seems a bit low, so it would be interesting to know if the authors have performed any sample size estimation and assessed the validity of their results according to the sample size.

The authors write, "After approval by an institutional ethics committee...». I would invite them to describe this specific part better, showing the name of the ethics committee, the protocol number, or stating how they managed data and informed consent.

I would invite the authors to assess better and describe the limitations of the research. There is a considerable selection bias, the sample size seems low, and more details about the questionnaire would be helpful. These aspects are beneficial to understand the external validity of their findings.

Author Response

Dear Reviewer, 

On behalf of all authors, I thank you for your attention and professionalism in reviewing our work. I have no doubt that the publication will become much better thanks to your work.

Prof. Gadi Segal, MD

Comment 1:

The authors describe that «we did a cross sectional study of user experience using questionnaires collected from physicians treating COVID-19 patients at Internal». I would invite them to make a better description of the questionnaires. If they were previously validated, or if they required translation or transcultural adaptation.

Answer 1:

The questionnaires were designed together with our epidemiology consultant as simple, Likert-based questionnaires following the relevant literature recommended principles. We added a description to the “methods” section with a main reference also inserted.

Comment 2:

The authors describe that «We collected twenty-five questionnaires and analysed them to ensure compliance with exclusion and inclusion criteria. Following this, only 18 questionnaires were included in the analysis». This number seems a bit low, so it would be interesting to know if the authors have performed any sample size estimation and assessed the validity of their results according to the sample size.

Answer 2:

We addressed the issue of “small groups” in our professional data analysis. We worked closely with a senior epidemiologist and our statistical presentation is both clear and sound: this is the reason for the addition of the “Cohen’s effect size” calculations.

Comment 3:

The authors write, "After approval by an institutional ethics committee...». I would invite them to describe this specific part better, showing the name of the ethics committee, the protocol number, or stating how they managed data and informed consent.

Answer 3:

The detailed number was added to the “methods” section. Also regarding the informed consent.

Comment 4:

I would invite the authors to assess better and describe the limitations of the research. There is a considerable selection bias, the sample size seems low, and more details about the questionnaire would be helpful. These aspects are beneficial to understand the external validity of their findings.

Answer 4:

The way of reducing error and increasing validity while investigating a small group was further detailed in the “methods” section and also mentioned once again in the “limitation” section of the manuscript.

Round 2

Reviewer 1 Report

The revised version of the work results indeed more homogeneous and the more prudent sentences makes it much more scientifically sound and definitively suitable to be published. 

Beside some minor typos, I will enlist below, there is however a small suggestion that I'd like to share with the authors: the results of the survey highlight that clinicians tends to prefer direct auscultation whenever the patient is accessible. Beside the usual "cultural" prudence in the adoption of novel technologies in the medical field, it could be interesting to read something about the concerns or the reasons for these opinions by clinicians. Is it a matter of responsibility? Does it refers to the quality of sounds? Does it regards the usability in broad sense (i.e. difficulty for the patient or caregiver to properly place the device) ? Is it the time requested to complete the task? Other relevant reasons ?

Minor typo or comments:

Row 40: these -> similar

Row 260: hipnotyze -> hypotise

Author Response

Reviewer 1

Comment 1: The revised version of the work results indeed more homogeneous and the more prudent sentences makes it much more scientifically sound and definitively suitable to be published. 

Answer 1: we thank you very much for appreciating the work invested. Indeed, your review significantly improved this manuscript.

Comment 2: Beside some minor typos, I will enlist below, there is however a small suggestion that I'd like to share with the authors: the results of the survey highlight that clinicians tends to prefer direct auscultation whenever the patient is accessible. Beside the usual "cultural" prudence in the adoption of novel technologies in the medical field, it could be interesting to read something about the concerns or the reasons for these opinions by clinicians. Is it a matter of responsibility? Does it refers to the quality of sounds? Does it regards the usability in broad sense (i.e. difficulty for the patient or caregiver to properly place the device) ? Is it the time requested to complete the task? Other relevant reasons ?

Answer 2: since the questionnaire we used did not address the reasons for preferences, we could only speculate. We added a remark concerning the need for such questions to be asked in future studies (at the end of the “conclusions” paragraph).

Comment 3: Row 40: these -> similar

Answer 3: corrected!

Comment 4: Row 260: hipnotyze -> hypotise

Answer 4: this word was not found. Most probably already corrected.

Reviewer 2 Report

The authors have addressed my concerns

Author Response

We thank you very much